# From Tokens to Actions: Discrete Flow Matching for Visuomotor Policy Learning

## Abstract

Although actions in the physical world are inherently continuous, representing them in a discrete space can unlock stability, efficiency, and multimodality in policy learning. We present Discrete Flow Matching Policy (DFMP), a novel method that learns continuous robot actions in a discrete space using score-based generative modeling. DFMP formulates action generation as a Continuous-Time Markov Chain, learning transition probabilities over action tokens. Through this, DFMP unifies three desirable properties: (i) stable optimization through flow-matching objectives, (ii) multimodal behavior modeling via probabilistic branching between tokens, and (iii) fast inference. To bridge continuous control with discrete representations, we systematically study tokenization schemes and analyze their trade-offs, proposing the optimal approach for real world robot policies. We thoroughly evaluate DFMP across many challenging simulated manipulation benchmarks and two real-world robot deployments, showing that our approach provides not only strong task performance, but also better scalability and robustness compared to existing continuous-space methods. These results position DFMP as a new, principled approach to efficient, robust, and multimodal visuomotor policy learning, advancing the integration of discrete generative modeling into real-world robotics. Videos and code are provided on the project page https://dfm-policy.github.io.

## 1 Introduction

Humans acquire dexterous motor skills not by solving physical equations, but by learning from a vast amount of sensory experience: watching, practicing, and generalizing across contexts (Lake et al., 2017). This ability to distill patterns from large-scale data and reuse them in multiple tasks is what makes human intelligence so adaptable. A similar story has unfolded in vision and language, where large models trained on massive datasets have shown striking generalization, emergent reasoning, and transfer across domains (Radford et al., 2021; Brown et al., 2020). In robotics, it has become increasingly clear that the same principle holds: scaling visuomotor policy learning with large and diverse datasets is key to building robots that can move beyond brittle, task-specific solutions toward flexible, general-purpose control (Brohan et al., 2022; 2023a;b).

Despite great progress that has been made in learning robotic policies from demonstration data (Zhang et al., 2022; Zeng et al., 2021), designing architectures that can fully capture the richness of human demonstrations remains an unsolved challenge. Compared to generating text or images, where multimodality often manifests at a coarse level (e.g. alternative word choices, objects, or colors), robot actions are much finer temporally and spatially. For example, a grasp can succeed from several distinct end-effector poses, or a door opening action can be executed at slightly different angles, but averaging across these high-frequency variations yields invalid or unstable actions (Lynch et al., 2020).

In our experiments, we find that continuous generative models often struggle with this kind of multimodality. Diffusion-based policies can represent diverse behaviors, but the iterative denoising process blurs high-frequency details and results in slow inference (Chi et al., 2023). Flow matching provides efficient, stable training, but tends to collapse onto a single dominant trajectory, ignoring

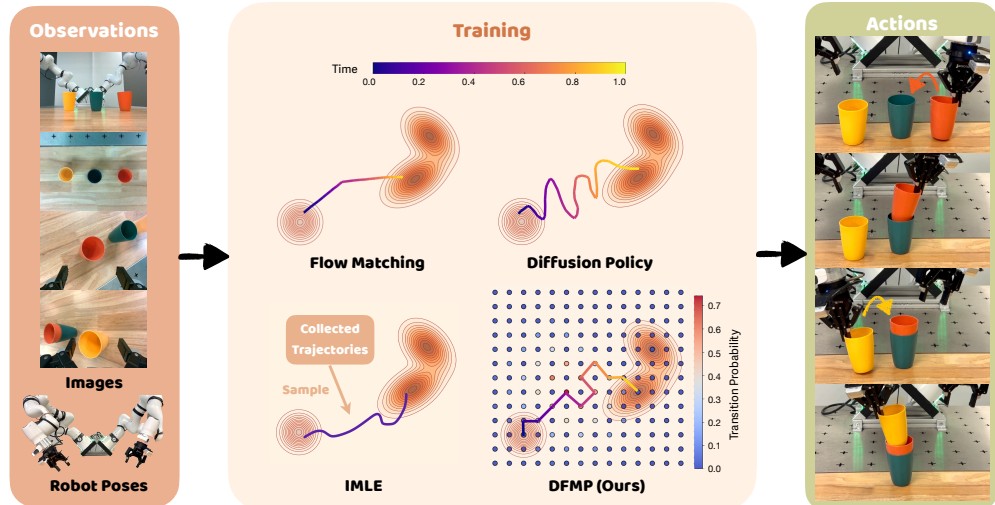

Figure 1: **DFMP** models discrete robot actions as transitions in a Continuous-Time Markov chain, capturing multimodality with efficient training and fast inference, enabling robust real-world manipulation.

alternative action variants (Lipman et al., 2023b). Nearest neighbors approaches are data efficient and allow for sampling precise multiple modes of behavior; however, they are brittle due to low generalizability (Duan et al., 2017). Thus, we ask the following question: **How can we design a policy class that is both stable and efficient, yet still preserves the fine-grained multimodality inherent in robot demonstrations?**

The transformation of actions into tokens alters this geometry: Instead of averaging, the probability mass can branch explicitly between different tokens, allowing for the preservation of multiple valid action variants without collapse. To this end, we introduce a novel state-of-the-art policy architecture: Discrete Flow Matching Policy (DFMP). Inspired by discrete generative modeling, DFMP treats robotic actions as tokens and models generation as a Continuous-time Markov Chain (CTMC). Our proposed architecture is inherently multimodal and, by design, addresses issues with previous continuous generative modeling approaches. This formulation merges the stability of flow-based training with the expressiveness of discrete multimodal models, while enabling fast inference through non-iterative sampling.

Our contributions are threefold: (1) we present a novel policy architecture that leverages discrete flow matching (Gat et al., 2024) for robot action learning, formulating action generation as a token-based continuous-time process; (2) we systematically study different tokenization schemes and analyze their trade-offs for representing continuous actions; and (3) we demonstrate across eight simulated manipulation benchmarks and two real-world robot deployments that DFMP consistently outperforms or matches state-of-the-art baselines, achieving faster convergence, stronger multimodality, and more reliable generalization. By bridging discrete generative modeling and robotics, we show that continuous actions can be treated as tokens, opening new directions for efficient multimodal policy learning. Videos and results are available at: https://dfm-policy.github.io

## 2 RELATED WORK

**Learning From Demonstrations** A widely used approach in robotic learning is to imitate expert actions, collected through kinesthetic teaching, teleoperation, or scripted demonstrations (Schaal, 1999; Ijspeert et al., 2002; 2003; Argall et al., 2009; Ziebart et al., 2010; Bain & Sammut, 1995; Jang et al., 2021). Behavior Cloning (BC) directly maps observations to actions (Pomerleau, 1989; Torabi et al., 2018), and has been widely applied in domains such as autonomous driving and robotic manipulation. Despite its simplicity, BC methods often struggle with complex human demonstrations, requiring large datasets and suffering from mode averaging and poor generalization (Laskey et al., 2017; Mandlekar et al., 2021a; Zhao et al., 2020). They are also prone to compounding errors due to covariate shift (Ross et al., 2011; Kelly et al., 2019; Osa et al., 2018), motivating ex-

tensions such as Dataset Aggregation (DAgger) (Ross et al., 2011), noise-injection methods like DART (Laskey et al., 2017), hierarchical approaches (Lynch et al., 2019), and recent large-scale dataset-driven variants (Kalashnikov et al., 2021; Jang et al., 2021).

**Generative Modeling for Policy Learning**    More recently, approaches leverage generative models to capture the richness of expert demonstrations. Common choices include energy-based models (Florence et al., 2022) or diffusion-based ones (Ho et al., 2020; Chen & et al., 2022; Chi et al., 2023; Janner et al., 2022). While these have been able to capture complex human demonstrations (Zhao et al., 2023; Dasari et al., 2023) and achieve strong results in manipulation tasks (Shafiullah et al., 2023), they have costly inference procedures, making deployment more challenging (Black et al., 2023). Flow matching methods (Lipman et al., 2023b; Albergo & Vanden-Eijnden, 2023; Liu et al., 2023) offer a more efficient alternative, directly learning continuous normalizing flows. Such policies (Shao et al., 2023; Wu et al., 2023b; Kim et al., 2023b) require less tuning and allow for faster inference. However, flow-based approaches can collapse into singular modes more often (Tong et al., 2024). Sample matching approaches (Li & Malik, 2019; Rana et al., 2025) are more data-efficient and can reproduce precise modes; however, they generalize poorly when deployed in the real world.

**Discrete Generative Models and Tokenization**    Generative modeling over discrete distributions has produced very successful models with strong language and vision capabilities (Yang et al., 2023; Rombach et al., 2022). Representing data as tokens enables stable training and branching across multimodal outcomes (van den Oord et al., 2017; Gat et al., 2024). Similarly, in robotics, tokenized action representations, for example VQ-VAEs (Shafiullah et al., 2022a; Wu et al., 2023a) or simple binning (Brohan et al., 2023a; Ahn et al., 2022; Reed et al., 2022a), have been used for trajectory modeling. More recently, Vision-Language-Action models (VLAs) (Brohan et al., 2023a; Kim et al., 2023a; Huang et al., 2024; Kim et al., 2024) leverage and predict in a common token space of language, pixels, and actions. They are complementary to generative BC policies (Chen et al., 2023; Team, 2023; Reed et al., 2022b), and thus face similar drawbacks. Therefore, our work fills the gap by combining the stability and efficiency of flow-based objectives with the multimodal flexibility of discrete action spaces, as well as a specialized architecture to handle discrete multimodal prediction.

# 3 METHODS

In this section, we present our approach, **Discrete Flow Matching Policy (DFMP)**, which reframes visuomotor policy learning as generative modeling over a tokenized action space. We leverage a specialized architecture as well as a training approach, inspired by discrete normalizing flows. Figure 2 summarizes the overall framework.

## 3.1 DISCRETE FLOW MATCHING POLICY FORMULATION

**CTMC**    We formulate the generation process from source tokens to target tokens as a **Continuous-Time Markov Chain (CTMC)**, charactarized by a probability transition kernel $p_{t+h|t}$, which specifies the probability of transitioning from timestep $t$ to $t + h$:

$$p_{t+h|t}(y \mid x) = \mathbb{P}(X_{t+h} = y \mid X_t = x) = \delta(y, x) + h\, u_t(y, x), \quad (1)$$

where $\delta(y, x) = 1$ if $y = x$ and 0 otherwise. The term $u_t(y, x)$ represents the *transition velocity*, which indicates the instantaneous rate at which probability mass flows between tokens as a function of time $t$, with $0 \leq t \leq 1$.

**Discrete Probability Paths**    In the general flow matching framework (Lipman et al., 2023a), the objective is to construct a probability path $p_t$ that continuously transforms a source distribution $p$ into a target distribution $q$. In the discrete setting, we couple source tokens $\mathbf{x}_0 \sim p$ and target tokens $\mathbf{x}_1 \sim q$, and construct the *marginal probability path* using samples drawn from the joint distribution $\pi(x_0, x_1)$. The conditional formulation is then obtained by marginalization:

$$p_t(x) = \sum_{x_0, x_1 \in \mathcal{V}^d} p_t(x \mid x_0, x_1)\, \pi(x_0, x_1), \quad \text{where} \quad p_t(x \mid x_0, x_1) = \prod_{i=1}^{N} p_t\big(x^i \mid x_0^i, x_1^i\big), \quad (2)$$

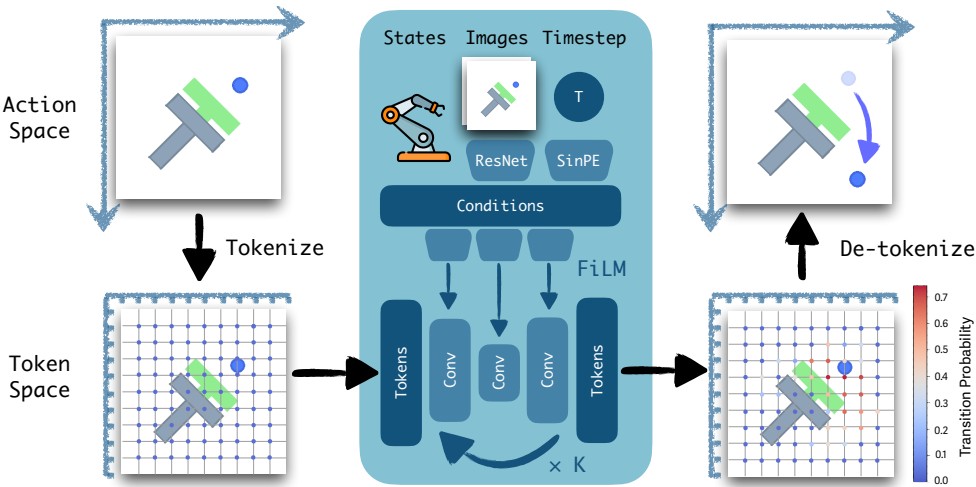

Figure 2: Overview of our approach DFMP: we discretize actions, learn transitions with flow matching, and reconstruct continuous trajectories.

with $p_t(x^i \mid x_0^i, x_1^i)$ denoting a time-dependent conditional distribution over the token pair $(x_0, x_1)$. This path satisfies the boundary conditions $p_0(x^i \mid x_0^i, x_1^i) = \delta(x^i, x_0^i)$, $p_1(x^i \mid x_0^i, x_1^i) = \delta(x^i, x_1^i)$. In practice, we implement the conditional probability path using a quadratic scheduler $t^2$ as it provides a smoother start and sharper convergence:

$$p_t(x^i \mid x_0^i, x_1^i) = (1 - t^2)\, \delta(x^i, x_0^i) + t^2\, \delta(x^i, x_1^i). \tag{3}$$

**Discrete Velocity Field** Following Section 3.1, we interpret the generation process through the lens of CTMCs, where $u_t$ is referred to as the *discrete velocity field*. To ensure that the constructed path $p_t$ is feasible, $u_t$ must satisfy the Kolmogorov Equation:

$$\frac{\mathrm{d}}{\mathrm{d}t} p_t(y) = \sum_x u_t(y, x)\, p_t(x), \tag{4}$$

which states that the rate of change of probability mass at state $y$ equals the total inflow from all other states $x$. By substituting the conditional probability path in equation 3, we obtain the corresponding velocity field by solving equation 4:

$$u_t^i(y^i, x^i \mid x_0^i, x_1^i) = \frac{2t}{1 - t^2}\left[ \delta(y^i, x_1^i) - \delta(y^i, x^i) \right]. \tag{5}$$

This closed-form expression characterizes the instantaneous dynamics of probability mass transfer in the discrete setting and serves as the foundation for the subsequent policy learning framework.

## 3.2 IMPLEMENTATION FOR DFMP

Discrete Flow Matching (Gat et al., 2024) was originally proposed for generating discrete tokens from dummy or mask tokens for language modeling. To adapt this framework for action prediction, we introduce two key modifications. First, we discretize continuous actions from the Euclidean space $\mathbb{R}^d$ into a tokenized representation within the discrete space $\mathcal{V}^d$, where the vocabulary is defined as $\mathcal{V} = \{1, 2, \ldots, V\}$. After training, the generated token sequences are mapped back into continuous action vectors and executed. Second, the generation process must be conditioned on observations $O_t$ (raw visual inputs as well as proprioception). An overview of the proposed DFMP algorithm, incorporating these modifications, is presented in Figure 2.

### 3.2.1 TOKENIZERS

In this section, we discuss the various options for the discretization scheme used for DFMP.

*(1) Lift    (2) Can    (3) Square    (4) Tool Hang    (5) Transport    (6) Push-T    (7) Block Push    (8) Kitchen*

Figure 3: **Simulated Environments**. *Robomimic Benchmark* (Mandlekar et al., 2021b) (1-5), *Push-T* (Florence et al., 2021) (6), *UR3 Block Push* (Shafiullah et al., 2022b) (7) and *Franka Kitchen* (Gupta et al., 2019)(8).

**Bin** A bin-based tokenizer divides each action dimension $a^d \in [l^d, u^d]$ into $B^d$ equal-width intervals. Encoding maps $a^d$ to its bin index $k^d = \left\lfloor \frac{a^d - l^d}{u^d - l^d} \cdot B^d \right\rfloor$, and decoding uses the bin center. This method is extremely simple and computationally efficient, requiring no data-dependent training.

**Quantile** A quantile-based tokenizer partitions each action dimension by empirical quantiles so that each bin contains approximately the same number of samples. If $q_j^d$ are the quantile cut points, then $k^d = j$ if $q_j^d \le a^d < q_{j+1}^d$. This ensures balanced token frequencies, which often improves training stability and reduces class imbalance.

**KD-Tree** The KD-Tree tokenizer recursively partitions the joint action space $\mathbb{R}^d$ along coordinate axes at medians until $V$ leaves are formed. Each leaf corresponds to a token ID, $k = \text{leaf}(a)$ for $a \in \mathbb{R}^d$. Unlike independent per-dimension discretizers, this method captures cross-dimensional dependencies and balances token usage by construction.

**VQ-VAE** The VQ-VAE (van den Oord et al., 2018) learns a discrete codebook $\{e_k\}_{k=1}^V$ to quantize continuous actions in a data-adaptive manner. An encoder maps an action $a$ to a latent $z_e$, which is replaced by its nearest code $e_k$, and a decoder reconstructs $\hat{a} = D(e_k)$. It is trained with loss $\mathcal{L} = \|a - \hat{a}\|_2^2 + \beta \|\text{sg}[z_e] - e_k\|_2^2 + \|z_e - \text{sg}[e_k]\|_2^2$, where $\text{sg}[\cdot]$ is a stop-gradient operator.

### 3.2.2 DFMP TRAINING LOSS

To learn the velocity field introduced in equation 5, we parameterize a neural network $u_t^\theta$ with trainable parameters $\theta$. Because the velocity field must depend jointly on samples and observation data, we adopt the *Conditional Discrete Flow Matching* (CDFM) objective, which takes the form

$$\mathcal{L}_{\text{CDFM}}(\theta) = \mathbb{E}_{t, x_t \sim p_t | x_0, x_1, O_t} D_{x_t}\big(u_t(\cdot, x_t \mid x_0, x_1), u_t^\theta(\cdot, x_t \mid O_t)\big), \tag{6}$$

where $D_{x_t}(\cdot, \cdot)$ denotes the Bregman divergence. Lipman et al. (2024) demonstrate that $\mathcal{L}_{\text{CDFM}}(\theta)$ and $\mathcal{L}_{\text{DFM}}(\theta)$ produce identical gradients with respect to $\theta$. In practice, Bregman divergence has been observed to yield more stable convergence behavior than either KL divergence or cross-entropy loss in the discrete flow matching setting.

### 3.2.3 DFMP NETWORK ARCHITECTURE

For the implementation of $u_t^\theta$, we adopt a CNN-based U-Net architecture following the designs of Chi et al. (2023) and Rana et al. (2025). Unlike these prior approaches, which directly regress continuous robot actions $a \in \mathbb{R}^d$, our formulation in the discrete flow matching setting predicts a probability distribution over tokens $x \in \mathcal{V}^d$ drawn from a finite vocabulary. A softmax is then applied across all tokens, encouraging multimodality in the learned action distribution and improving the smoothness of generated actions.

For vision-based control tasks, we employ ResNet-18 (He et al., 2015) as an encoder to transform raw image sequences into compact latent embeddings. These embeddings serve as input to the policy network, which operates in a closed-loop receding-horizon control framework. At each time step $t$, the policy conditions on the most recent $T_o$ observation steps and predicts the next $T_p$ steps of future actions. The robot then executes $T_a$ actions before re-planning, which ensures both temporal consistency and adaptability in dynamic environments.

Figure 4: We perform thorough real-world evaluations, with significant amounts of variations. (a) Randomized cup positions in *Cup Stacking* task. (b) Randomized cup poses in *Cup Placing* task. (c) Training cup in *Cup Placing* task. (d) Unseen cups.

## 4 EXPERIMENTS

We systematically evaluate DFMP on both simulated and two different real-world robot platforms, and perform rigorous experiments and ablations. We aim to answer the following questions:

1) How does DFMP compare against state-of-the-art policy learning approaches?
2) In which tasks and domains does DFMP demonstrate the greatest advantages?
3) How do design decisions such as different tokenizers (Section 3.2.1) affect performance?

**Baselines** For experiments, we use the following baselines: **Diffusion Policy**: The vanilla Diffusion Policy introduced by Chi et al. (2023), trained using the DDPM framework. **Flow Matching**: A baseline that modifies the U-Net architectures of Chi et al. (2023) by incorporating the Flow Matching objective proposed in Lipman et al. (2023a). **IMLE**: A visuomotor policy learning method based on Implicit Maximum Likelihood Estimation, as proposed by Rana et al. (2025).

### 4.1 SIMULATION BENCHMARKS

| Multimodality | Task | HiPrec | IMLE | Diffusion Policy | Flow Matching | DFMP (Ours) |
|---|---|---|---|---|---|---|
| **Low** | Lift | | **1.00/1.00** | **1.00/1.00** | **1.00/1.00** | **1.00/1.00** |
| | Can | | 0.98/0.96 | 0.98/0.98 | **1.00/1.00** | 0.98/0.96 |
| | Square | ✓ | 0.62/0.60 | 0.84/0.84 | **0.88/0.84** | **0.88/0.86** |
| | Tool Hang | ✓ | **0.78/0.72** | 0.52/0.44 | 0.68/0.64 | 0.76/0.70 |
| | Transport | ✓ | 0.80/0.74 | 0.92/0.88 | 0.92/0.82 | **0.94/0.86** |
| **High** | Kitchen | | **1.00/0.98** | 1.00/0.96 | 0.92/0.86 | **1.00/0.98** |
| | Block Push | | 0.90/0.82 | **0.94/0.90** | 0.90/0.88 | **0.94/0.90** |
| | Push-T | ✓ | 0.96/0.90 | 0.98/0.96 | 0.94/0.88 | **1.00/0.94** |

Table 1: Simulated benchmark success rates in the format of max success rate / average of last 3 checkpoints, averaged across 50 trials. HiPrec: whether the task has a high precision requirement.

We evaluate the DFMP on 8 simulation tasks drawn from 4 diverse environments, including both state and vision observations, different precision requires, task horizons sensory inputs. (1) *Robomimic* (Mandlekar et al., 2021b) manipulation setup. It includes five tasks: *Lift*, *Can*, *Square*, *Tool Hang* , and *Transport*. Success is defined as placing objects at their designated target locations. (2) *Push-T* (Florence et al., 2021) involves a circular end-effector pushing a T-shaped block (both starting from random positions) to a fixed target. (3) *UR3 Block Push* (Shafiullah et al., 2022b), a task in a UR3 robot pushes two blocks to goal circles. Success is achieved when both blocks reach their goals. (4) *Franka Kitchen* (Gupta et al., 2019), a multi-task manipulation setup, featuring a 7-DoF Franka Panda arm interacting with seven kitchen objects. Success is measured by completing four distinct (long and short horizon) tasks sequentially. All tasks except *Push-T* are MuJoCo (Todorov et al., 2012) based.

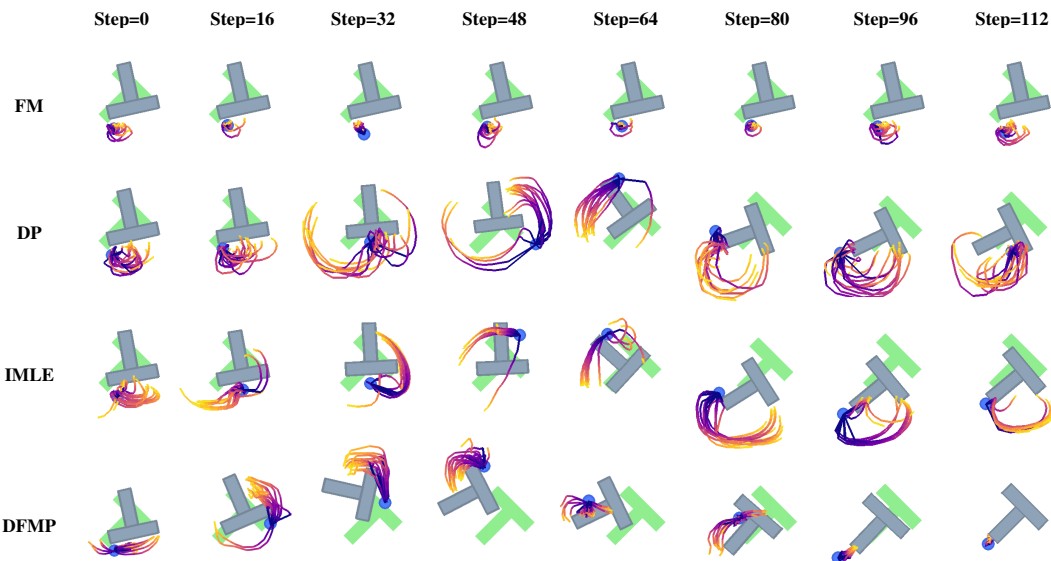

Figure 5: Trajectory visualizations comparing **Flow Matching (FM)** (Liu et al., 2023), **Diffusion Policy (DP)** (Chi et al., 2023), **IMLE** (Rana et al., 2025), and **DFMP** across different inference steps. We observe that DFMP is both fast and handles different behavior modes well.

**Analysis**   Table 1 reports the success rates across low- and high-multimodality tasks. For simpler low-multimodal tasks such as *Lift* and *Can*, all methods perform comparably and nearly achieve perfect success, indicating that these tasks do not strongly differentiate between approaches. However, for more challenging tasks that require high precision, such as *Square*, *Tool Hang*, and *Transport*, DFMP consistently achieves the best or near-best performance, demonstrating greater stability and robustness in fine-grained control.

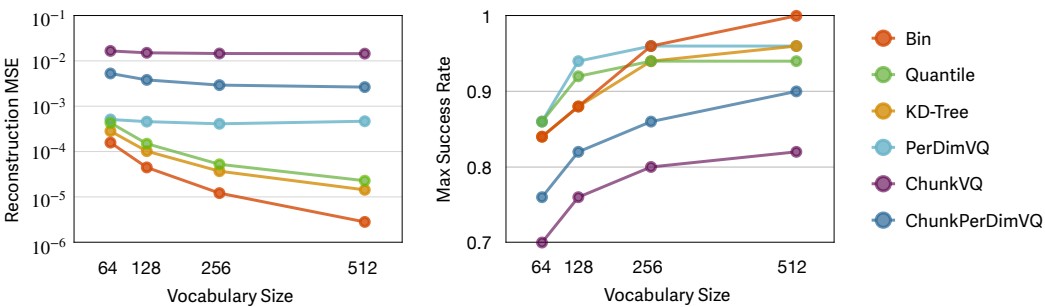

Figure 6: Reconstruction loss and success rate for different tokenizers across vocabulary sizes (Push-T task).

For tasks that require strong multimodality, we see that Implicit Maximum Likelihood Estimation (IMLE) struggles due to its reliance on samples in the existing dataset. Moreover, its training speed can be relatively slow, as shown in Figure 7. Diffusion Policy, on the other hand, is able to achieve similar results to DFMP but its iterative sampling process leads to long training times and slow inference. As shown in Figure 5, where we can see that Diffusion Policy is slower to converge compared to the DFMP, even when the same number of inference steps is assigned. This explains its delayed but steep convergence trend in Figure 7 and its instability in precision-demanding tasks (e.g. *Tool Hang*). Flow Matching, while being more efficient at test time, is not as multimodal as other approaches. This limitation is clearly visible in Figure 5: it collapses to a single mode and gets stuck in corner cases without recovering (Figure 5), despite achieving fast convergence as shown in Figure 7. On the other hand, DFMP not only completes task inference within fewer steps compared to Diffusion Policy, but also demonstrates diverse trajectories, avoiding the mode-collapse issue of

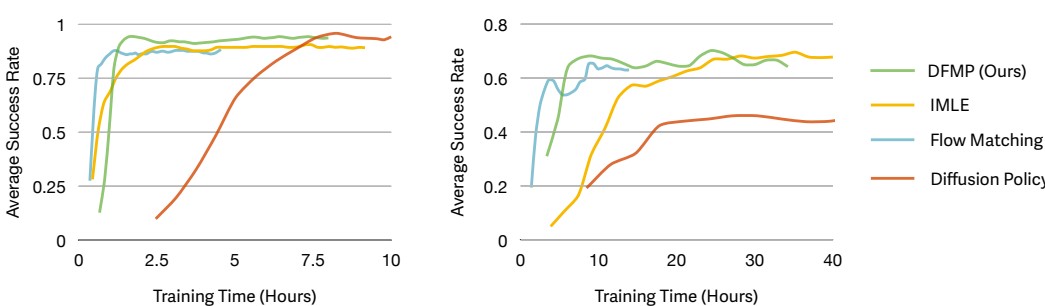

Figure 7: DFMP achieves faster convergence and higher success across *Push-T* and *Tool Hang*.

vanilla Flow Matching. As shown in Table 1, Figure 7, DFMP matches or surpasses baselines in most tasks, combining efficiency, robustness, and multimodality.

**Tokenizer Ablations**  We ablate and analyze different tokenization schemes in the *Push-T* task. As shown in Figure 6, this inevitably introduces loss of information. Larger vocabularies reduce reconstruction error and improve success, but also increase model size and training overhead. Among the different schemes, Quantile and KD-Tree are better for small and medium size vocabularies, with KD-Tree being more balanced overall. PerDimVQ, which learns a separate codebook for each action dimension, performs surprisingly well with small vocabularies but saturates at larger ones. ChunkVQ, which compresses an entire action chunk into a single token, is very compact but suffers from severe information loss. In contrast, ChunkPerDimVQ mitigates some loss by assigning one token per dimension within each chunk. Bin achieves the best accuracy with a large vocabulary, which is attractive when accuracy is a priority but with a higher inference cost. Overall, we find performance is maximized when accuracy is desired. We thus employ Bin for subsequent experiments.

## 4.2 REAL WORLD TASKS

**Experiment Setup**  We evaluated DFMP in the real world for two tasks in two hardware configurations shown in Figure 8. This first task is *Cup Stacking*, which is performed on a bimanual Franka system with four cameras (top, front, two wrists). The robot must sequentially stack cups: the left arm places the orange cup into the green cup, followed by the right arm stacking the yellow cup on top. The initial positions of the cup are randomized within a range (Figure 4 (a)), and the task is long-horizon and precision-critical, as even minor misalignment can cause the stack to collapse. The second task *Cup Placing* is conducted on a single-arm UR5 with three cameras (top, front, wrist). The robot is required to pick up a randomly placed cup on

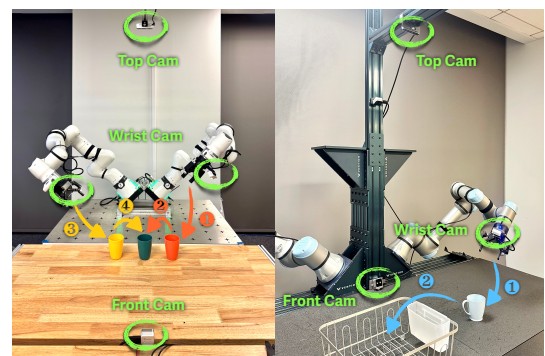

Figure 8: Real-world experiments setup. **Left:** *Cup Stacking* on Franka. **Right:** *Cup Placing* on UR5.

the table (Figure 4 (b)) and place it into a dish rack. This task is multi-modal, since there are multiple valid ways to grasp the cup. We also use some unseen cup (Figure 4 (d)) to test policy generalization ablity. For each task, we collected 120 demos at 30Hz. The evaluation is also run at 30Hz. All training is done on a single H200. For DFMP, IMLE and Flow Matching, training can be finished within 10 hours. For Diffusion Policy, the evaluation loss is converged after 20-hour training. For the sake of fairness, we compare all methods which are fully trained, and early stop Diffusion Policy (terminate after 10-hour training).

**Analysis**  In our real-world experiments, we evaluate all methods on two manipulation tasks, *Cup Stacking* and *Cup Placing*, as well as on an unseen object for generalization testing. The results reveal several interesting trends.

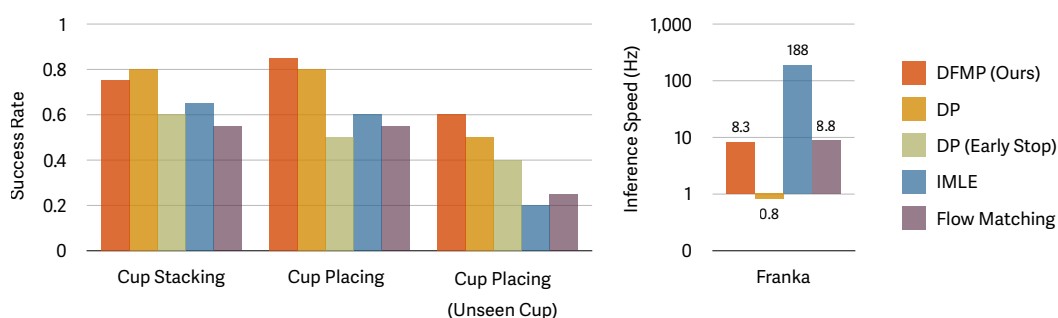

Figure 9: Comparison of success rate and inference speed in real-world experiments. All results are averaged over 20 runs.

For *Cup Stacking*, which requires precise and fine-grained control, DFMP performs slightly worse than a fully trained Diffusion Policy. This gap can be attributed to the tokenization process in DFMP, which inevitably introduces some information loss. Nevertheless, DFMP still significantly outperforms IMLE, Flow Matching, and an early-stopped Diffusion Policy trained with the same compute budget as DFMP, highlighting a better efficiency-performance trade-off for our approach.

For *Cup Placing*, a highly multimodal task, both DFMP and Diffusion Policy exhibit strong success rates, demonstrating their ability to capture diverse grasping strategies. In contrast, IMLE and Flow Matching often collapse to a single grasping mode during evaluation, failing to capture multimodality. To assess generalization, we evaluate policies on an unseen cup not present in the training set. Consistent with our earlier analysis in Section 4.1, IMLE shows a sharp drop in the success rate, confirming its limited generalizability. By comparison, DFMP maintains stable performance, benefiting from its discrete flow formulation.

In terms of inference speed, Diffusion Policy is the slowest due to its iterative denoising process, while IMLE achieves the fastest inference as a single-step method. DFMP lies in between: although not as fast as IMLE, it is nearly two orders of magnitude faster than Diffusion Policy while preserving multimodality and robust generalization. Importantly, the inference speed of flow matching-based approaches, although still below the control frequency of 30Hz, remains practically feasible since the policy predicts $T_p$ steps that can be executed over $T_a$ steps. This design ensures stable execution while maintaining sufficient reactivity.

## 5 CONCLUSION AND LIMITATION

We presented DFMP, a discrete flow matching framework that brings score-based generative modeling into the tokenized action space for robot learning. Across eight simulated manipulation tasks and two real-world evaluations, DFMP demonstrates balanced performance: it converges faster than Diffusion Policy, generalizes more reliably than IMLE, and captures multimodality more effectively than vanilla Flow Matching. These results highlight the benefits of combining flow-based efficiency with a discrete representation of robot actions.

Nevertheless, DFMP has limitations. Tokenization inevitably introduces some information loss, which may affect tasks requiring extremely fine-grained control. In addition, the vocabulary size must be predefined, creating a trade-off between training efficiency and representational capacity. Smaller vocabularies save computation but risk losing expressiveness, while larger vocabularies increase overhead. A promising direction for future work is to design adaptive or learned tokenization strategies tailored to robot action spaces, enabling compact yet information-rich encodings that further improve generalization and efficiency.

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

# A APPENDIX

## A.1 DERIVATION OF EQ. 5

Starting from Eq. 4 applied to the $i$-th coordinate, we have

$$\frac{d}{dt} p_t^i(y^i \mid x_0^i, x_1^i) = \sum_{x^i} u_t^i(y^i, x^i \mid x_0^i, x_1^i) \, p_t^i(x^i \mid x_0^i, x_1^i).$$

Substitute the conditional path from Eq. 3:

$$p_t^i(x^i \mid x_0^i, x_1^i) = (1 - t^2) \, \delta(x^i, x_0^i) + t^2 \, \delta(x^i, x_1^i).$$

Compute its derivative:

$$\frac{d}{dt} p_t^i(y^i \mid x_0^i, x_1^i) = 2t \left[ \delta(y^i, x_1^i) - \delta(y^i, x_0^i) \right].$$

Plug this into the Kolmogorov equation:

$$2t \left[ \delta(y^i, x_1^i) - \delta(y^i, x_0^i) \right] = u_t^i(y^i, x_0^i \mid x_0^i, x_1^i) \, (1 - t^2) + u_t^i(y^i, x_1^i \mid x_0^i, x_1^i) \, t^2.$$

Impose the CTMC generator constraint (rows of the generator sum to zero):

$$u_t^i(y^i, x_0^i \mid x_0^i, x_1^i) + u_t^i(y^i, x_1^i \mid x_0^i, x_1^i) = 0, \quad \Rightarrow \quad u_t^i(y^i, x_1^i \mid x_0^i, x_1^i) = -u_t^i(y^i, x_0^i \mid x_0^i, x_1^i).$$

Solve for $u_t^i$:

$$u_t^i(y^i, x_1^i \mid x_0^i, x_1^i) = \frac{2t}{1 - t^2} \left[ \delta(y^i, x_1^i) - \delta(y^i, x_0^i) \right].$$

This yields Eq. (5) in per-coordinate form:

$$u_t^i(y^i, x^i \mid x_0^i, x_1^i) = \frac{2t}{1 - t^2} \left[ \delta(y^i, x_1^i) - \delta(y^i, x_0^i) \right].$$

## A.2 TASK SPECIFICATIONS

| Task | ActDim | ObsDim | #Robots | #Objects | #Demo | #Cam | #Steps | UseImg | HiPrec | HiModal |
|---|---|---|---|---|---|---|---|---|---|---|
| **Simulation Benchmark** | | | | | | | | | | |
| Lift | 7 | 11 | 1 | 1 | 200 | 2 | 400 | Yes | No | No |
| Can | 7 | 11 | 1 | 1 | 200 | 2 | 400 | Yes | No | No |
| Square | 7 | 11 | 1 | 1 | 200 | 2 | 400 | Yes | Yes | No |
| Transport | 14 | 22 | 2 | 3 | 200 | 4 | 700 | Yes | No | No |
| Tool Hang | 7 | 11 | 1 | 2 | 200 | 2 | 700 | Yes | Yes | No |
| Push-T | 2 | 4 | 1 | 1 | 200 | 1 | 1000 | Yes | Yes | Yes |
| UR3 Block Push | 2 | 4 | 1 | 2 | 1000 | - | 1000 | No | No | Yes |
| Kitchen | 9 | 60 | 1 | 7 | 656 | - | 1000 | No | No | Yes |
| **Realworld Benchmark** | | | | | | | | | | |
| Cup Stacking | 16 | 16 | 2 | 3 | 120 | 4 | 1000 | Yes | Yes | No |
| Cup Placing | 7 | 7 | 1 | 2 | 120 | 3 | 1000 | Yes | No | Yes |

Table 2: ActDim: action dimensions, ObsDim: state observation dimensions, #Robots: number of robots, #Objects: number of objects, #Demo: number of demonstrations in dataset, #Cam: number of cameras, #Steps: max number of rollout steps in evaluation, Use Img: whether the task uses image as observations, HiPrec: whether the task has a high precision requirement, HiModal: whether the task is high multimodality.

| Task | VocabSize | BatchSize | $T_o$ | $T_p$ | $T_a$ | #FlowIters | #DiffIters | #Sample | EMA |
|---|---|---|---|---|---|---|---|---|---|
| **Simulation Benchmark** | | | | | | | | | |
| Lift | 256 | 128 | 2 | 16 | 8 | 10 | 100 | 20 | 0.75 |
| Can | 256 | 128 | 2 | 16 | 8 | 10 | 100 | 20 | 0.75 |
| Square | 128 | 128 | 2 | 16 | 4 | 10 | 100 | 20 | 0.75 |
| Transport | 512 | 128 | 2 | 16 | 8 | 10 | 100 | 20 | 0.75 |
| Tool Hang | 512 | 64 | 2 | 16 | 4 | 10 | 100 | 20 | 0.75 |
| Push-T | 512 | 128 | 2 | 32 | 8 | 10 | 100 | 20 | 0.75 |
| UR3 Block Push | 64 | 64 | 2 | 32 | 8 | 10 | 100 | 20 | 0.75 |
| Kitchen | 128 | 128 | 2 | 16 | 8 | 10 | 100 | 20 | 0.75 |
| **Realworld Benchmark** | | | | | | | | | |
| Cup Stacking | 256 | 16 | 4 | 32 | 8 | 10 | 100 | 20 | 0.75 |
| Cup Placing | 256 | 16 | 4 | 32 | 8 | 10 | 100 | 20 | 0.75 |

Table 3: VocabSize: vocabulary size for tokenization, BatchSize: batch size used for training, $T_o$: observation horizon, $T_p$: horizon of predicted actions, $T_a$: horizon of executed actions, #FlowIters: number of flow iterations in Flow Matching and DFMP, #DiffIters: number of diffusion iterations in Diffusion Policy, #Sample: number of samples per condition in IMLE, EMA: smoothing factor in Exponential Moving Average.

### A.3 TRAINING HYPERPARAMETERS

### A.4 NETWORK PARAMETERS

- **Diffusion step embedding dimension:** 256
- **Vision feature dimension:** 512
- **U-Net layer size:** [256, 512, 1024]
- **Convolution kernel size:** 5
- **Learning rate in Adam optimizer:** 1e-4
- **Weight decay:** 1e-6

### A.5 HARDWARE DETAILS

We evaluated our policies on two real-world robot platforms, a Franka Emika Panda and a UR5 manipulator, using an Intel RealSense D450 camera to capture image observations. Demonstration data is collected with the Gello system (Wu et al., 2024), and the robots are operated under position control.

### A.6 LLM USAGE

We used large language models (LLMs), in particular ChatGPT, solely as an auxiliary tool during manuscript preparation. Specifically, the LLM was used for polishing the writing, checking LaTeX syntax for mathematical formulas, searching for relevant references, and assisting in minor code debugging. The LLM was not used to generate research ideas, design experiments, analyze results, or draft technical content. All scientific contributions, methodology, and experimental findings presented in this work are the original work of the authors.

