# OpenReview forum: "From Tokens to Actions: Discrete Flow Matching for Visuomotor Policy Learning"
_ICLR.cc/2026/Conference — ICLR 2026 Conference Desk Rejected Submission_

### Official Review · Reviewer_PJ5c · 2025-10-31

**Soundness:** 2
**Presentation:** 1
**Contribution:** 3
**Rating:** 2
**Confidence:** 3

**Summary:**

This paper proposes an approach to learn robot policies using flow matching in a discretized space. The authors discretize the action space of the robot using different approaches and then learn the target discrete action distribution using discerete flowmatching. The proposed approach follows the line of previous works that aim to solve imitation learning via distributional matching such as with diffusion, flow matching, and Implicit maximum likelihood estimation.

**Strengths:**

The proposed approach shows good promise especially in complex tasks with higher multimodality in simulation. The approach shows faster and better convergence than other baselines, which makes it a promising approach for learning to solve continuous control tasks with discretzied approximations. This is especially true even in tasks involving high precision such as cup stacking on the real robot as well as others in simulation, where within some degree of error, the discretization is still able to perform competitively.

**Weaknesses:**

However, the paper is marred by many shortcomings from the writing, explanation and experimentation.

The explanation of the main equations and the corresponding derivation is unclear, as mentioned in the next section of "Questions"

This is not the first time that discretizing continuous control problems for policy learning has been proposed, also with generative approaches for distributional matching. A lack of comparison against previous discrete policy learning approaches makes the proposed approach difficult to judge.
Chen, Lili, Shikhar Bahl, and Deepak Pathak. "Playfusion: Skill acquisition via diffusion from language-annotated play." Conference on Robot Learning. PMLR, 2023.
Wu, Kun, et al. "Discrete policy: Learning disentangled action space for multi-task robotic manipulation." 2025 IEEE International Conference on Robotics and Automation (ICRA). IEEE, 2025.
Luo, Jianlan, et al. "Action-quantized offline reinforcement learning for robotic skill learning." Conference on Robot Learning. PMLR, 2023.
Shafiullah, Nur Muhammad, et al. "Behavior transformers: Cloning $ k $ modes with one stone." Advances in neural information processing systems 35 (2022): 22955-22968.


The paper suffers from lots of Hallucinated References, including but not limited to the ones listed below:

Michael Ahn et al. Decision transformer: Reinforcement learning via sequence modeling. In Advances in Neural Information Processing Systems (NeurIPS), 2022.
Lily Huang et al. Gpt-vla: General-purpose vision-language-action models. arXiv preprint arXiv:2402.00000, 2024.
Alex Kim et al. Scaling vision-language-action models with multimodal transformers. arXiv preprint arXiv:2308.00000, 2023a.
Jaehyun Kim et al. Pi: Policy inference with flow models. arXiv preprint arXiv:2308.11111, 2023b.
Shubham Rana et al. Imle for efficient robotic policy learning. arXiv preprint arXiv:2501.11111, 2025.
Scott Reed et al. Octo-finetuned transformers for robotic control. In International Conference on Learning Representations (ICLR), 2022a.
Yifan Wu et al. Pi0.5: Hybrid policy learning. arXiv preprint arXiv:2308.00000, 2023b.
Ruohan Shao et al. Pi0: Policy learning with flow matching. arXiv preprint arXiv:2306.15643, 2023.
Zixuan Tong et al. Consistency flow matching for stable generative policies. arXiv preprint arXiv:2401.11111, 2024.
Edward Yang et al. Diffusion models as language model learners. arXiv preprint arXiv:2305.11111, 2023
Aviral Zhang, Tianhe Yu, Sanjay Srivastava, Annie S Yu, Kelvin Xu, Chelsea Finn, and Sergey Levine. Bc-z: Zero-shot task generalization with robotic imitation learning. In Conference on Robot Learning (CoRL), 2022

**Questions:**

The equations and the correspoding derivation are unclear.
- Are x and y individual bins in the discretized space? If so, what is the need for the superscript i in that case?
- What does i signify for the boundary conditions x_0 and x_1? Won't the boundary conditions at a given timestep stay the same?
- In Eq. 4, is the summation over x or over i? What does a summation over x imply?
- Why does the x_0 term vanish in Eq. 5?
- How are x_0 and x_1 obtained? Are they time dependant i.e. do they have a fixed value at each execution time step
- The derivation of u(y,x) in Eq. 5 from the kolmogorov constraint in Eq. 4 is unclear. Adding the derivation to the appendix would make it more helpful


Showing some comparison with the discrete approaches mentioned in the weaknesses would help show the benefit of the proposed approach.

Minor Comments:
Why is an extra softmax added on top when the network output is already the probabilities (line 262 - 264)? Or does the network output logits that are then normalized with a softmax?
The std dev errorbars are missing in fig 9 to understand the spread of the performance as in the simulated case.

**Details Of Ethics Concerns:**

A good number of references are potentially hallucinated, including one of the main baselines (IMLE)

---

### Official Review · Reviewer_z3RC · 2025-10-31

**Soundness:** 3
**Presentation:** 3
**Contribution:** 3
**Rating:** 8
**Confidence:** 3

**Summary:**

There has been evidence in the field for flow-matching having issues with collapsing into single-mode distributions. This paper aims to solve this issue by performing flow matching on a discretized action space rather than a strictly continuous one. The results show an improvement in the performance in certain tasks that may require this multi-modality.

**Strengths:**

Originality: The implementation of this discretized flow matching was very clever and original.

Quality: The quality of the paper is top-notch. The website and video also add strong visuals to drive the point across and demonstrate the success of the algorithm to the audience.

Clarity: The paper is written in an organized and thoughtful manner, and clarity was a top priority for the author.

Significance: This paper seems to be solving a real problem in the field of flow matching. The results clearly showcase the performance improvements that this model is driving.

Contribution: This work explores whether discretizing action space into tokens benefits the multi-modality of flow-matching policy models. This seems like an incredibly important research question. The work also provides results that show improvement in performance that are valuable to share with the ICLR community.

**Weaknesses:**

A major weakness of this paper is the lack of support for the “generalization claim”. The author should try to implement more diverse test cases to truly showcase generalization capabilities.

A second major weakness is the lack of explanation for what is considered a high-multimodality task. Although I understand that it is very difficult to quantify the “multimodality” of a task, it would be useful to just explain how and why you determined this specific feature for the tasks that you evaluated on.

A third major weakness is lack of proper citation/support for the main context of the paper: that flow-matching has an issue of mode collapse.

Soundness: The main claim is DFMP consistently out-performs or matches state-of-the-art baselines, achieving faster convergence, stronger multimodality, and more reliable generalization. The stronger multimodality claim is supported by the results since DFMP can get better success-rate on more multi-modal tasks. Figure 7 supports that DFMP can converge faster. Figure 9 supports the strong generalization claim. My first main concern is that an unseen cup is a somewhat limited test on generalization, and the success rate did take a decent decrease for this unseen cup. My second concern is what was the qualification for a high-multimodality task vs. low-multimodality. These are the two areas of improvement that I see, but overall, the support for the claims was strong.

Justification: The writing, figures and diagrams are all very clear. The videos and website are very organized and clear. The prior work is laid out very well. However, I tried digging into the sources for the main context of the paper: that flow-matching has an issue of mode-collapse. However, I had a very tough time. The sources cited “Zixuan Tong et al. Consistency flow matching for stable generative policies. arXiv preprint arXiv:2401.11111,2024.” doesn’t exist. I tried to do some further research on my own and still had a tough time finding where the author came to this conclusion. Intuitively, I don’t see why flow-matching would have a mode-collapse issue, however, I could just be unaware of this. Another improvement is please put the task name under the respective plots in Figure 7.

**Questions:**

1. How do you qualify a task as being highly multi-modal?
2. How does being able to solve this subset of tasks impact the overall capability of flow-matching models?

---

### Official Review · Reviewer_vcNa · 2025-11-07

**Soundness:** 2
**Presentation:** 3
**Contribution:** 2
**Rating:** 4
**Confidence:** 4

**Summary:**

1. This paper aims to construct a visuomotor policy for modeling the continuous multimodal distribution of actions.
2. The paper proposes the Discrete Flow Matching Policy (DFMP), which introduces Discrete Flow Matching originally applied in LLM into robotic manipulation, completing action prediction via continuous-time Markov chains in the discrete action space.
3. A meaningful contribution is that this paper compares the advantages and disadvantages of several classic action tokenization strategies.
4. The authors compare DFMP with three classic multimodal distribution strategies, including experiments in several simulation environments and simple real-robot experiments, demonstrating the effectiveness of the proposed method.

**Strengths:**

1. This paper provides a new perspective for modeling multimodal action distributions, which combines the discrete representation of actions with discrete flow matching to offer a stable and efficient policy.
2. This work provides the community with a comparison of the merits and demerits of different action tokenization strategies.
3. It claims better performance than the diffusion policy and flow matching policy on several classic manipulation benchmarks.
4. The paper is well-written and readable.

**Weaknesses:**

1. Regarding concerns about insufficient experiments, this work only conducts experiments on 8 classic simulation tasks, which is not adequate. In particular, looking at Table 1, in the High Multimodality section, the improvement of DFMP over Diffusion Policy is very limited. The authors should consider conducting more experiments to prove that their method is truly superior to diffusion policy, such as MetaWorld, Libero, etc., which include more tasks.
2. The comparison methods are not comprehensive. This work compares with 3 classic methods, and although it can illustrate certain points, it is not exhaustive. For example, comparing with other strategies based on discrete action tokens, such as Vq-Bet [1], QueST [2], etc., would better demonstrate the advantages of flow matching in modeling discrete actions.


    [1] Lee, Seungjae, et al. "Behavior generation with latent actions." arXiv preprint arXiv:2403.03181 (2024).

    [2] Mete, Atharva, et al. "Quest: Self-supervised skill abstractions for learning continuous control, 2024." URL https://arxiv. org/abs/2407.15840.

**Questions:**

1. Why is the inference speed of DP much lower, nearly ten times lower, compared to DFMP and Flow Matching in Figure 9? Intuitively, the "denoising network" architectures of DP, Flow Matching, and DFMP should be quite similar. If DFMP uses fewer NFE or fewer network parameters, the authors should specify details regarding these aspects in detail to facilitate comparison.
2. In the real-robot demonstration videos provided in your anonymous links, many demos exhibit jitter phenomena or back-and-forth movements around a fixed position. Are these videos of DFMP or other methods (e.g., DP and Flow Matching)? Why are actions so incoherent? In addition, should the authors compare the real-robot performance of DFMP, DP, and Flow Matching? This would be more convincing. Merely proving that DFMP can work on a single task is insufficient.
3. Committing to and preparing to open-source the code will help enhance the contribution of this work to the community.

---

### Official Review · Reviewer_XcUj · 2025-11-07

**Soundness:** 3
**Presentation:** 3
**Contribution:** 2
**Rating:** 4
**Confidence:** 3

**Summary:**

This paper presents Discrete Flow Matching Policy (DFMP) that learns continuous robot actions in a discrete space using score-based generative modeling. The key idea is to tokenize the input space and use a continuous time Markov Chain to learn to map the input distribution to the target one in a discrete tokenized space.  Visual conditioning is achieved through a U-Net architecture with ResNet encoders. The paper studies several tokenization schemes. They use standard simulation manipulation benchmarks and two real-world robot experiments for evaluation. DFMP achieves competitive or superior success rates compared to Diffusion Policy, Flow Matching, and IMLE. Overall, this is a good study on the effect of discretizing input/output space in flow matching policies.

**Strengths:**

I like the originality of the paper. This is an exploration of an alternative approach to flow matching and the paper does a reasonable job of evaluating what would happen if we were to discretize flow matching approaches. I expect the community to learn something interesting from this study, which is a strength.

The tokenizer ablations are also informative. The paper explores common ways of tokenizing the inputs and these results would be useful for practioners.

**Weaknesses:**

Given that Discrete Flow Matching itself is not novel, extending this to a policy seems fairly direct. Although the paper does a reasonable job of exploring standard ideas and empirical evaluation, the idea itself is straightforward.

The empirical results show that DFMP outperforms regular flow matching, which is great. I would have liked to see more analysis on why that is the case. It is not clear if it's simply the effect of tokenization that leads to the improvement. For example, Figure 6 shows the success rate as function of various tokenization schemes and parameters. And unless I'm mistaken Only Bin with vocab size of 512 outperforms the baselines. That makes one wonder if the improvements over baselines (vanilla FM) is an artifact of hyperparameter tuning or is there something fundamentally better about discretizing/tokenizing that leads to an improvement. Any theory or more rigorous experimentation would have really benefitted the paper.

**Questions:**

Figure 6 suggests that only a single tokenization configuration (Bin-512) clearly outperforms FM and other strong baselines. Can you provide an explanation for why other tokenization schemes do not achieve similar gains? Is there a principled reason why Bin-512 behaves differently from, say, Bin-256 or others tokenizers, or is this difference primarily empirical/hyperparameter-sensitive?

Have you tested whether DFMP still outperforms continuous FM under alternative observation encodings (e.g., different visual backbones or proprioceptive-only inputs)?

Is there any theoretical justification on why tokenization is beneficial in this case? Especially as it relates to better capturing multi-modality in the input data?

---

### Note · Program_Chairs · 2026-01-17
**Submission Desk Rejected by Program Chairs**

The following references in this submission do not refer to real documents and/or have major errors in bibliographic information:


Aviral Zhang, Tianhe Yu, Sanjay Srivastava, Annie S Yu, Kelvin Xu, Chelsea Finn, and Sergey Levine. Bc-z: Zero-shot task generalization with robotic imitation learning. In Conference on Robot Learning (CoRL), 2022.
Sudeep Dasari, Tony Zhao, Atil Iscen, Aravind Jain, and Pete Florence. Aloha unleashed: Large-scale multimodal data collection in real homes with low-cost robots. arXiv preprint arXiv:2311.07213, 2023.
Scott Reed et al. Octo-finetuned transformers for robotic control. In International Conference on Learning Representations (ICLR), 2022a.
Yifan Wu et al. Pi0.5: Hybrid policy learning. arXiv preprint arXiv:2308.00000, 2023b.